# Performance of some early-maturing groundnut (*Arachis hypogaea* L.) genotypes and selection of high-yielding genotypes in the potato-fallow system

**Gangadhara K, Ajay BC** \*, **Praveen Kona, Kirti Rani, Narendra Kumar, S. K. Bera** \*

ICAR-Directorate of Groundnut Research, Junagadh, Gujarat, India

\* berask67@yahoo.co.in (BSK); ajaygpb@yahoo.co.in (ABC)

**Data Availability Statement:** All relevant data are within the manuscript and its Supporting Information files.

## Abstract

India imports the most edible oils because domestic demand exceeds production. Horizontally expanding groundnut production in non-traditional areas especially in the potato-paddy rice-fallow system is possible for increasing production and it requires trait-specific cultivars. Only 1% of oilseeds are grown in non-traditional regions. Nine interspecific groundnut derivatives were tested in potato-fallow system at Deesa, Gujarat, and Mohanpura, West Bengal, and non-potato fallow areas in Junagadh during Kharif 2020 to examine their performance and adaptability. Genotype-by-environment (G×E) interaction significantly affected pod yield and its components in the combined ANOVA. "Mean vs. stability" showed that the interspecific derivative NRCGCS 446 and variety TAG 24 were the most stable and valuable genotypes. GG 7 yielded more pods in Junagadh, whereas NRCGCS 254 yielded more in Mohanpur. Low heritability estimates and strong G×E interaction for flowering days showed complicated inheritance and environmental effects. The shelling percentage was significantly correlated with days to 50% blooming, days to maturity, SCMR, HPW, and KLWR, demonstrating negative connections between maturity, component characteristics, and seed size realisation.

## Introduction

Though India achieved self-sufficiency in oilseeds during the early 1990s, since 1994, there has been a continuous increase in oil imports. India imports around 15 million tons of edible oils worth approximately 900 million dollars, which accounted for 40% of the agricultural imports bill and 3% of the overall import bill of the country in 2019 [1]. With the efforts of the Technology Mission on Oilseeds (TMO), oilseed production increased from 10.8 m.t. in 19 m ha of the area during 1985–86 to 25 mt in 27 m ha of the area during 1998–1999 [2]. The increase in oilseed production was possible due to the availability of quality seeds of improved cultivars, advanced technology services, and the price support policy. But with the ever-increasing demand for food legumes and oilseeds due to the continuous rise in the population and

**Funding:** The work presented in this article is a contribution from institute funded research projects and no external grants were involved.

**Competing interests:** The authors have declared that no competing interests exist.

competition from other remunerative crops, it is necessary to increase oilseed production both horizontally and vertically in the country. Groundnut is a major oilseed crop in India and can be grown in three seasons viz., *Kharif*, pre-*rabi*, *rabi*, or summer in different parts of India [3]. However, in India, groundnut is mainly cultivated as a *Kharif* crop under rainfed conditions with an average yield of 1635 kg/ha. However, groundnut cultivation carried out in pre-*rabi*, *rabi*, or summer seasons is more remunerative due to higher productivity (1922 kg/ha) under irrigated condition, better crop management, and low environmental pressure by biotic stresses.

It is estimated that about 0.7 to 1 million ha additional areas [2] could be used for groundnut cultivation as a pre-*rabi*/*rabi*/summer crop either after *Kharif* paddy or pre-*rabi*/*rabi* potatoes in lands, which are otherwise kept fallow. These areas are mostly located in Eastern and North-Eastern states of India, which account for 6.08 lakh ha of the total *Kharif* area (Triennial average area from 2017–18 to 2019–20), [2] besides potato-fallow in Uttar Pradesh with 5.65 lakh ha area (Triennial average area from 2017–18 to 2019–20), [2]. These areas are reffered to as non-traditional areas for groundnut. In these non-traditional areas *Kharif* paddy or pre-*rabi*/*rabi* potatoes are grown under rainfed conditions using long duration varieties and cropping duration extends upto last week of November. With the lack of rains from November and non-availability of irrigation source it is not possible to take up long duration crops. Available residual moisture could be utilized by growing short duration groundnut varieties.

Rice–fallow cropland areas are the regions of paddy rice cultivation during the *Kharif* growing season that were kept fallow in the *rabi* season due to lack of irrigation, late harvesting of high-yielding long–duration rice varieties, moistures stress at sowing time during the *rabi* season, early withdrawal of the monsoon, waterlogging, excessive moistures in November or December, and crop damage caused by stray cattle and blue bulls [4]. Rice fallow (~11.7 million ha) is a rice-based production system, mainly concentrated in the eastern states, i.e., Chhattisgarh, Jharkhand, Upper Assam, Bihar, eastern Uttar Pradesh, Odisha, and West Bengal [5–7]. Soil moisture stress at the time of fallow crop sowing results in low plant stand [6]. Even if the crop is established well with the utilization of residual soil moisture, the lack of winter rain during the reproductive stage often leads to the complete failure of crops [8]. One of the major constraints for utilization of these rice fallows is the lack of short duration (80–90 days) varieties, which can fit into the sowing windows of major crops. Interspecific derivatives represent a good source of valuable alleles conferring various economically important traits. A considerable number of interspecific derivatives have been developed and further used in breeding for groundnut improvement at the ICAR-Directorate of Groundnut Research, Junagadh [9–20] Present study is aimed at identifying stable high yielding and early maturing genotypes from among interspecific derivatives for use in potato-fallow system.

## Materials and methods

### Experimental details

Nine interspecific derivatives/mutants/breeding lines were selected out of 450 breeding lines based on the number of days to fifty percent flowering of short-day varieties [21]. The experiment involving nine breeding lines, along with three short-duration check varieties (Table 1), was conducted at the ICAR-Directorate of Groundnut Research, Junagadh, India during *Kharif* crop season (June to October) in 2017, 2018, and 2019 in a medium black calcareous (12% CaCO3), clayey, Ustochrept soil, with the minimum temperature ranging from 18.2 to 28.1˚C (February to May) and the maximum temperature ranging from 29.5 to 35.6˚C (Table 2). The experiment was arranged in a randomized complete block design (RCBD) with three replications. Each accession was planted in a single row of 3 m in length, with a spacing

**Table 1. The available information on the pedigree of nine interspecific derivatives and check varieties used in the present study.**

| SN | Interspecific derivative | Habit Group | Pedigree |
|----|--------------------------|-------------|----------|
| 1 | NRCG CS 254 | SB | J 11 × A. stenosperma |
| 2 | NRCG CS 292 | SB | J 11 × Black testa mutant |
| 3 | NRCG CS 313 | SB | J 11 × A. duranensis |
| 4 | NRCG CS 330 | SB | J 11 × Black testa mutant |
| 5 | NRCG CS 400 | SB | GG 2 mutant |
| 6 | NRCG CS 404 | SB | GG 2 mutant |
| 7 | NRCG CS 445 | SB | J 11 × A. pusilla |
| 8 | NRCG CS 446 | SB | J 11 × A. pusilla |
| 9 | NRCG CS 62 | VB | CT7-1 × SB XI |
| 10 | Chico (ICG 476) | SB | An early-maturing germplasm line from U S A |
| 11 | Pratap Mungphali 1 (PM1) | SB | ICGV 86033 × ICG-2214) |
| 12 | TAG 24 | SB | Selection from TGE 2 × TGE 1 |

of 60 cm between rows × 10 cm between plants. Standard agricultural practices and plant protection measures were adopted for healthy crop production. Data on days to maturity (DM), days to flower bud initiation (DFI), days to 50% flowering (DFF), SPAD chlorophyll meter readings (SCMR), pod yield per plant (PYPP; g), hundred pod weight (HPW; g), shelling percentage (SP; %), kernel length (KL; mm), kernel width (KW; mm), and kernel length/width ratio (KLWR) were recorded from five randomly selected plants of each genotype under each replication. Days to flower bud initiation and days to 50% flowering were recorded as the number of days from planting to first flowering and the date of planting to reach flowering in 50% of the plants of a genotype, respectively. The surrogate trait for water use efficiency, SCMR, was measured 60 days after planting using a Minolta handheld portable SCMR meter (SPAD-502 plus Minolta, Tokyo, Japan). Other data were recorded at harvest.

Based on the initial 3-year evaluation of nine breeding lines at the ICAR-DGR Junagadh, two selected breeding lines (NRCGCS 252 and NRCGCS 446) and four check cultivars (TAG 24, TG 37 A, GG 7, and JL 501) were further evaluated at three locations (Deesa and Junagadh in Gujarat and Mohanpur in West Bengal) during summer 2020. The experiments were conducted at Deesa and Mohanpur after harvesting potato crops. Approximately 71352 and 68611 ha of fallow potato areas are found in Deesa and Mohanpur (Average area between the periods 2016–17 and 2019–20, [2], respectively, which can be used for the succeeding groundnut crop

**Table 2. Weather parameters recorded at Junagadh during the *Kharif* season in 2017–2019.**

| Month | June | | | July | | | August | | | September | | | October | | |
|-------|------|------|------|------|------|------|--------|------|------|-----------|------|------|---------|------|------|
| Year | 2017 | 2018 | 2019 | 2017 | 2018 | 2019 | 2017 | 2018 | 2019 | 2017 | 2018 | 2019 | 2017 | 2018 | 2019 |
| Max Temperature (˚C) | 37 | 37.1 | 36.6 | 30.9 | 30.7 | 33.9 | 31.3 | 3 0.5 | 30.4 | 32.4 | 32.1 | 31.1 | 36.5 | 37.4 | 33 |
| Min Temperature (˚C) | 26.6 | 28.1 | 27.1 | 25.4 | 25.8 | 26.5 | 24.6 | 25 | 25.3 | 24.5 | 23.5 | 25.1 | 22.9 | 21.5 | 23 |
| Mean Temperature (˚C) | 31.8 | 32.6 | 31.9 | 28.1 | 28.3 | 30.2 | 28 | 27.7 | 27.8 | 28.5 | 27.8 | 28.1 | 29.7 | 29.5 | 28.8 |
| Relative humidity (%) | 69 | 65 | 69 | 85 | 85 | 75 | 84 | 83 | 90 | 79 | 71 | 88 | 51 | 47 | 65 |
| Wind speed (Km/h) | 23.9 | 12.5 | 9.8 | 25 | 9.3 | 10.7 | 25.3 | 9.3 | 7.5 | 27.4 | 5.2 | 4.1 | 20.2 | 2.7 | 3.1 |
| Evaporation (mm) | 6.8 | 8.2 | 6.8 | 2.4 | 2.9 | 4.9 | 3 | 2.9 | 2.5 | 3.9 | 4.3 | 1.5 | 5.9 | 5.5 | 3.6 |
| Total rainfall (mm) | 147.8 | 7.4 | 138 | 330.5 | 641.9 | 228 | 282.6 | 88.6 | 393 | 43.5 | 51.5 | 678 | 0 | 0 | 42.2 |
| Total rainy days | 10 | 1 | 7 | 18 | 14 | 10 | 10 | 5 | 13 | 4 | 0 | 21 | 0 | 0 | 3 |

cultivation with minimum resources since potato is a crop grown in high input conditions, and residual nutrients available in fallow potato fields meet the major nutrient requirements of the succeeding groundnut crop. Deesa (a part of the Banaskantha district in Gujarat) is characterized by 'Ustalf' soil type with the minimum temperature within the ranges of 13.4 to 27.2˚C (February to May) and the maximum temperature ranging from 31.6 to 42.2˚C. Mohanpur, a part of the West Medinipur district in west Bengal, is characterized by 'orthid' soil type covering 45% of the area and 'Udupts' soil type covering the remaining 55% of the area. Temperatures in this location range from 16.5 to 27.0˚C (February to May) as minimum and 30.0 to 36.8˚C (February to May) as maximum. Experiments were laid out in randomized complete block design (RCBD) with three replications in all three locations. Standard agronomic practices and plant protection measures were adopted for healthy crop production. Data for pod yield and its related traits were recorded in experiments.

## Statistical analyses

The correlation [22], heritability [23], and variability parameters [24] were estimated. The stability and pod yield of genotypes were estimated over three years using the Additive Main Effects and Multiplicative Interaction (AMMI) model. AMMI stability Index (ASI) for each genotype was calculated [25] with the Agricolae package [26] in R version 4.1.0 [27]. Simultaneous selection indices (SSI) for yield and stability were calculated [28] to identify stable and high-yielding genotypes. The "mean versus stability" biplot was created in the R package GGE-BiplotGUI [29].

## Results and discussion

### Preliminary yield evaluation in Junagadh

The combined analysis of variance (ANOVA) showed that mean squares from genotype and environment obtained for pod yield per plant and other studied component traits were significant, except for kernel length/width ratio, indicating significant variation present among the genotypes under study, which offers great scope for improvement of traits through breeding line selection. Similar results have also been reported by earlier workers as well [30–33], for pod yield and its related characters. G×E interactions were significant for pod yield, days to flower bud initiation, days to 50% flowering, hundred pod weight, and kernel weight (Table 3), suggesting that genotypes respond differently from season to season, which is in agreement with the results of earlier studies [34].

Mean values for pod yield and its component traits (pooled over three years) are presented in Table 4 and season wise performance of genotypes for pod yield and component traits is presented in S1 Table. The pod yield and its component traits varied over the years. Days to first flowering ranged from 22 days for genotype NRCGCS 62 to 25 days for genotype PM 1, with a mean value of 24 days. As breeding was performed to select early-maturing lines, there was only three days difference between the two extreme genotypes. Such a narrow range was also observed for days to 50% flowering and days to maturity, with the former ranging from 25 to 28 days. Days to maturity in total ranged from 97 days (for NRCGCS 254 and NRCGCS 404) to 101 days (for NRCGCS 40), with a mean value of 98 days. SCMR ranged from 21.8 (NRCGCS 330) to 31.3 (NRCGCS 254), with a mean value of 28.45. Pod yield per plant ranged from 4.4 g (NRCGCS 62) to 13 g (NRCGCS 254), with a mean value of 7 g. The hundred pod weight ranged from 55 g (NRCGCS 40) to 82 g (NRCGCS 254), with a mean value of 71 g. Shelling percent ranged from 58% (NRCGCS 40) to 71% (Chico), with a mean value of 66%. Kernel length varied between 9 mm (NRCGCS 330) to 13 mm (NRCGCS 446), with a mean value of 11.1 mm, and kernel length to width ratio was within the ranges of 1.5 mm (NRCGCS

**Table 3. ANOVA for pod yield, SCMR, and maturity characteristics in the study conducted in Junagadh in 2017–2019.**

| EFFECT | Df | DFI | DFF | DM | SCMR | PYL |
|---|---|---|---|---|---|---|
| GEN | 11 | 4.97** | 5.5** | 19.44** | 58.4** | 41.12** |
| ENV | 2 | 59.53** | 60.73** | 183.26** | 1124.9** | 13.12** |
| REP(ENV) | 6 | 3.1** | 2.94** | 10.97 | 12.6 | 2.17 |
| GEN x ENV | 22 | 3.36** | 5.41** | 8.84 | 7.9 | 8.76** |
| RESIDUAL | 66 | 0.86 | 0.81 | 6.67 | 5.60 | 3.04 |
| CV | | 3.90 | 3.40 | 2.60 | 8.30 | 25.00 |
| L.S.D.(0.05) | | 0.87 | 0.84 | 2.43 | 2.23 | 1.64 |
| EFFECT | Df | HPW | SP | KL | KW | KLWR |
| GEN | 11 | 1169.6** | 143.6** | 12.3** | 2.12** | 0.21** |
| ENV | 2 | 3006.0** | 2221.6** | 635.7** | 218.3** | 0.03 |
| REP(ENV) | 6 | 110.0** | 25.5 | 0.3 | 0.29 | 0.02 |
| GEN x ENV | 22 | 93.7** | 20.6 | 1.2** | 0.41 | 0.05 |
| RESIDUAL | 66 | 35.40 | 23.30 | 0.60 | 0.26 | 0.04 |
| CV | | 8.30 | 6.80 | 7.00 | 8.00 | 11.70 |
| L.S.D. (0.05) | | 5.59 | 4.23 | 0.73 | 0.47 | 0.20 |

DFI-Days to first flower bud initiation; DFF-Days to 50% flowering; DMT-Days to maturity; HPW-Hundred pod weight (g); KL-Kernel length (mm); KW-Kernel width (mm); KLWR-Kernel length to width ratio; PYPP-Pod yield per plant; SCMR-SPAD chlorophyll meter readings; SP-Shelling percentage

40) to 2.08 mm (NRCGCS 446), with a mean value of 1.78 mm. Breeding lines (genotypes) NRCGCS 254 and NCRGCS 446 showed higher pod yield and larger seed size as compared to check (TAG 24) on average over three years. Two breeding lines, NRCGCS 446 (high average pod yield) and NCGCS 404 (moderate pod yield), and the check variety TAG 24, were found to be the most stable genotypes. Three breeding lines (NRCGCS 404, NRCGCS 254, and NRCGCS 330), on par with TAG 24, seemed to be early-maturing genotypes.

## Heritability estimation and correlation analysis

Analysis of genotypic coefficient variance (GCV), phenotypic coefficient variance (PCV), heritability, and genetic advance for different traits are given in Table 5. The phenotypic coefficient of variance (PCV) showed a slightly higher value than the genotypic coefficient of variance

**Table 4. The average performance of genotypes in terms of pod yield, SCMR, and maturity characteristics at Junagadh between 2017–2019.**

| S N | GEN | DFI (days) | DFF (days) | DM (days) | SCMR | PYLP (g) | HPW (g) | SP (%) | KL (mm) | KW (mm) | KLWR |
|---|---|---|---|---|---|---|---|---|---|---|---|
| 1 | NRCG CS 254 | 23 | 26 | 97 | 31.3 | 13 | 82.3 | 68.0 | 11.5 | 6.3 | 1.9 |
| 2 | NRCG CS 292 | 24 | 26 | 98 | 27.7 | 7 | 71.6 | 69.3 | 11.1 | 6.2 | 1.8 |
| 3 | NRCG CS 313 | 23 | 26 | 98 | 28.5 | 7 | 74.9 | 68.2 | 11.5 | 6.8 | 1.7 |
| 4 | NRCG CS 330 | 23 | 26 | 97 | 21.8 | 5 | 58.5 | 66.9 | 9.6 | 6.1 | 1.6 |
| 5 | NRCG CS 40 | 23 | 27 | 101 | 28.9 | 6 | 78.2 | 58.2 | 11.7 | 6.3 | 1.9 |
| 6 | NRCG CS 404 | 23 | 26 | 97 | 29.2 | 5 | 55.2 | 71.4 | 9.4 | 6.2 | 1.5 |
| 7 | NRCG CS 445 | 24 | 27 | 98 | 30.3 | 7 | 78.6 | 64.4 | 12.2 | 7.0 | 1.8 |
| 8 | NRCG CS 446 | 23 | 26 | 100 | 30.4 | 8 | 87.4 | 61.1 | 13.6 | 6.7 | 2.1 |
| 9 | NRCG CS 62 | 22 | 25 | 98 | 28.5 | 4 | 56.8 | 63.0 | 9.8 | 5.3 | 1.9 |
| 10 | Chico | 24 | 27 | 97 | 26.2 | 7 | 62.1 | 70.5 | 10.7 | 5.8 | 1.9 |
| 11 | PM 1 | 25 | 28 | 100 | 30.7 | 6 | 65.5 | 64.9 | 11.0 | 6.3 | 1.7 |
| 12 | TAG 24 | 24 | 26 | 97 | 27.9 | 7 | 84.7 | 68.9 | 11.1 | 6.9 | 1.6 |
| | LSD (0.05) | 0.9 | 0.8 | 2.4 | 2.2 | 1.6 | 5.6 | 4.2 | 0.7 | 0.5 | 0.2 |

**Table 5. Estimation of genetic parameters for pod yield, SCMR, and maturity characteristics in the study conducted in Junagadh in 2017–2019.**

|         | DFI  | DFF  | DM   | SCMR  | PYLP  | HPW    | SP    | KL    | KW   | KLWR  |
|---------|------|------|------|-------|-------|--------|-------|-------|------|-------|
| Min     | 22   | 25   | 97   | 21.82 | 4     | 55     | 58    | 9.43  | 5.26 | 1.52  |
| Max     | 25   | 28   | 101  | 31.31 | 13    | 87     | 71    | 13.60 | 6.97 | 2.08  |
| Mean    | 24   | 26   | 98   | 28.45 | 7     | 71     | 66    | 11.10 | 6.31 | 1.78  |
| VG      | 0.18 | 0.01 | 1.17 | 5.60  | 3.60  | 119.50 | 13.66 | 1.22  | 0.19 | 0.02  |
| VGXE    | 0.83 | 1.53 | 0.72 | 0.75  | 1.90  | 19.40  | 0.10  | 0.20  | 0.05 | 0.002 |
| VP      | 0.55 | 0.61 | 2.16 | 6.48  | 4.59  | 129.80 | 15.90 | 1.37  | 0.24 | 0.02  |
| GCV     | 1.77 | 0.38 | 1.10 | 8.31  | 27.10 | 15.39  | 5.60  | 9.53  | 6.91 | 7.95  |
| PCV     | 3.16 | 2.97 | 1.50 | 8.95  | 30.61 | 16.05  | 6.04  | 10.54 | 7.76 | 8.87  |
| h2 (bs) | 33   | 2.0  | 55   | 86    | 79    | 92     | 86    | 90    | 81   | 76    |
| GA      | 0.5  | 0.03 | 1.6  | 4.5   | 3.5   | 21.6   | 7.1   | 2.2   | 0.8  | 0.2   |
| GAM     | 2.1  | 0.1  | 1.7  | 15.9  | 50.0  | 30.3   | 10.7  | 19.4  | 12.8 | 13.5  |

(GCV) for the majority of traits, indicating that the traits under study were less strongly influenced by the environment. Similar results on groundnut were also reported earlier [35, 36]. The highest PCV (27.1%) and GCV (30.61%) values were found for pod yield per plant which is in agreement with earlier findings [35, 36]. The highest GCV and PVC values indicate the presence of high variability for pod yield, whereas moderate GCV (15.39%) and PCV (16.05%) values were recorded for the hundred pod weight. Low GCV (9.53%) and moderate PCV (10.54%) values were achieved for kernel length. Low GCV and PCV values were reported for days to flower bud initiation (1.77% and 3.16%), days to 50% flowering (0.38% and 2.97%), days to maturity (1.10% and 1.50%), SCMR (8.31% and 8.95%), shelling percent (5.60% and 6.04%), kernel width (6.91% and 7.76%), and kernel length to width ratio (7.95% and 8.87%), respectively, indicating lower variability existing among the genotypes for these traits. Low variability for these traits is also justified by the fact that the genotypes evaluated were all early-maturing genotypes.

Heritability is a good indicator of the transmission of traits from parents to progeny. Heritability is classified as low (below 30%), medium (30%-60%), and high (above 60%). Estimates of heritability help plant breeders use diversity in genetic resources to select superior genotypes. Therefore, high heritability estimates help effectively select any particular character. SCMR, pod yield per plant, hundred pod weight, shelling percent, kernel length, kernel width, and kernel length/width ratio had high heritability values. High heritability estimates for yield related traits and SCMR are in agreement with the earlier findings [37]. Moderate heritability was observed for days to maturity (55%), whereas days to flower bud initiation and days to 50% flowering showed low heritability; these results are in agreement with the findings of previously published reports [38]. Low heritability and genetic advance as percent of mean for days to flowering and days to maturity indicated the influence of environmental factors and polygenic inheritance of traits that are responsible for maturity.

The genetic advance is an useful indicator for making effective selection from the base population. GA and GAM [23] were classified as low (0–10%), moderate (10.1–20%), and high (> 20%). In the present study, the genetic advance was low for all the traits except for 100-kernel weight (21.6%). High genetic advance as percent of mean and also high heritability was recorded for pod yield per plant and 100-kernel weight, indicating the predominance of additive gene action for these traits. This shows that selection will be effective in improving these traits. High heritability with moderate genetic advance as percent mean was recorded for shelling percent, kernel length, kernel width, and kernel length/width ratio. Moderate heritability and low genetic advance were reported for days to flower bud initiation and days to maturity.

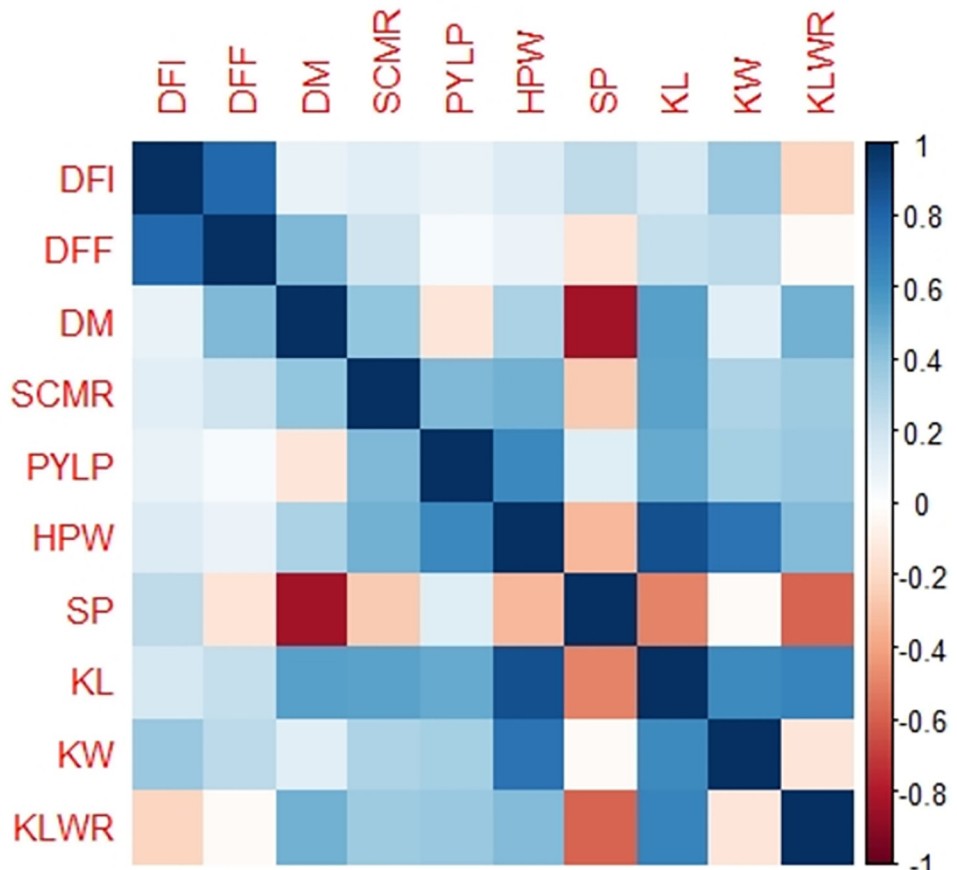

**Fig 1. Correlation plot showing relationships between pod yield and its component traits.**

Moreover, low heritability and influences of environment on days to flowering and days to maturity were also reported [39].

Pod yield showed a significant positive correlation with SCMR, but it was negatively correlated with days to maturity. Shelling percentage showed a significant negative correlation with days to 50% flowering, days to maturity, SCMR, HPW, and KLWR, suggesting a negative relationship between maturity duration and realization of seed size in groundnut (Fig 1).

### Evaluation of two interspecific derivatives and four varieties in the potato-fallow system

Pod yield and shelling percent of groundnut genotypes varied among locations, indicating possible crossover GE interactions. Differential performance was also observed among genotypes in potato-fallow in Deesa in Gujarat and Mohanpur in West Bengal and non-potato fallow in Junagadh, Gujarat (Fig 2). Selection of genotype is difficult when the performance of genotypes varies across different locations, and to overcome this problem and thus, select stable high-yielding genotypes, stability models such as AMMI and GGE biplots are being adopted extensively.

Environmental influence on the performance of six genotypes across three locations was studied using the AMMI model, which partitions the total variation due to genotype, environment, and interaction effect of genotype and environment (GEI) (Table 6). For pod yield per

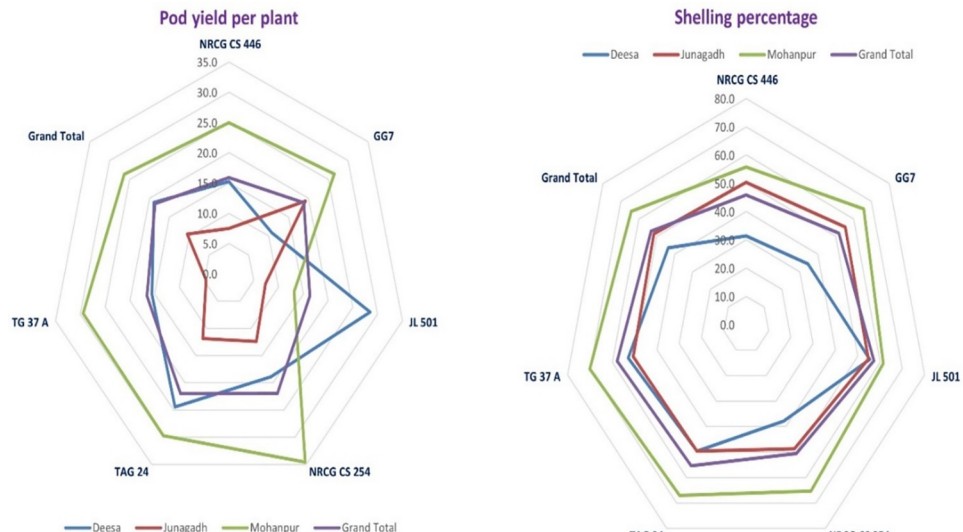

**Fig 2. Radar plot showing the differential performance of genotypes in potato fallow and non-potato fallow areas.**

plant, environment accounted for 44.04% of the total variation (maximum variation), followed by GEI (31.73%) and genotype (7.56%). For shelling percent, Environment, GEI, and genotype accounted for 56.96%, 17.08%, and 14.92% of the total variation, respectively. The contribution of E to the total variation was greater than that of G and GEI for pod yield per plant and shelling percent, indicating that the selection of genotypes based on multi-environment trials for indirect selection for target traits is the best strategy compared to selecting for a trait in one environment to obtain selection gains [40–47].

GEI was partitioned by the AMMI model into two principal components (IPCA1 and IPCA2), which together explained 100% of the total GEI interaction variation for pod yield per plant and shelling percent. For pod yield per plant, IPCA1 could explain 73.7% of the GEI variation, while IPCA2 explained 26.3% of the GEI variation. For shelling percent, IPCA1 explained 78.2%, but IPCA2 explained 21.8% of the GEI variation. Estimates of the average stability index (ASI) of 6 breeding lines for pod yield per plant and shelling percent are presented in Table 7. The lower the average stability index (ASI), the more stable the genotype. The stability alone may not serve breeding purposes, and instead, the search for genotypes with high yield and stability is required. Hence, simultaneous selection indices for yield and stability (SSI) were calculated. SSI incorporates both mean pod yield and stability in a single criterion. The low value of this parameter shows desirable genotypes with high mean pod yield and stability. Genotype NRCGS 446 was the most stable genotype with very low ASI values, whereas

**Table 6. Analysis of variance for AMMI model for pod yield per plant and shelling percentage of groundnut grown under the potato-fallow system.**

| Effect | df | Pod yield per plant | % Var | Shelling percentage | %Var |
|---|---|---|---|---|---|
| GEN | 5 | 47.78 | 7.56 | 127.8* | 14.92 |
| ENV | 2 | 695.8* | 44.04 | 1219.8** | 56.96 |
| GEN x ENV | 10 | 100.2** | 31.73 | 73.2* | 17.08 |
| REP(ENV) | 3 | 65.93 | 6.26 | 29.8 | 2.09 |
| RESIDUAL | 14 | 23.49 | 10.41 | 27.33 | 8.93 |
| IPCA1 | 6 | 182.7 | 73.7 | 146.02 | 78.20 |
| IPCA2 | 4 | 97.45 | 26.3 | 61.02 | 21.80 |

**Table 7. The AMMI stability Index (ASI) and simultaneous selection index for yield and stability (SSI) of 6 genotypes for pod yield per plant and shelling percentage.**

| Genotype | Pod yield per plant | | | | | Shelling percentage | | | | |
|---|---|---|---|---|---|---|---|---|---|---|
| | ASI | SSI | rASI | rY | means | ASI | SSI | rASI | rY | means |
| GG7 | 1.50 | 8 | 5 | 3 | 18.87 | 1.85 | 10 | 6 | 4 | 51.8 |
| JL501 | 2.71 | 11 | 6 | 5 | 16.33 | 1.67 | 7 | 5 | 2 | 57.2 |
| NRCGCS254 | 0.92 | 6 | 4 | 2 | 21.97 | 0.69 | 6 | 1 | 5 | 50.6 |
| NRCGCS446 | 0.25 | 7 | 1 | 6 | 15.88 | 1.32 | 10 | 4 | 6 | 45.8 |
| TAG24 | 0.38 | 3 | 2 | 1 | 22.00 | 0.98 | 5 | 2 | 3 | 55.4 |
| TG37A | 0.72 | 7 | 3 | 4 | 16.53 | 1.23 | 4 | 3 | 1 | 57.8 |

the check variety TAG 24 achieved the first rank in terms of SSI and was stable, with high yield. Genotype NRCGCS 254 was on par with TAG 24 in terms of stability and yield. For shelling percent, the check variety TG 37A ranked first with respect to SSI, and it was stable, with a high shelling percent.

The GGE analysis explains the two most important sources of variation, i.e., the genotype main effect (G) and the genotype × environment (GE) interaction effect, and it has been considered an effective method as the first two principal components (PC1 and PC2) of GGE explained 99% of the total variation. The GGE-biplot analysis is perfectly suitable for analysing multiple-environment trials, and it is based on the genetic correlation among the test environments (which-won-where pattern); the evaluation of environment carried out based on discrimination ability and representativeness, whereas the evaluation of genotype was made based on the mean performance and stability of genotypes across various environments. The GGE biplot graphically displays the G+GE of MET data in a way that facilitates visual variety evaluation and mega-environment identification [48, 49]. The which-won-where pattern of MET data is the polygon view whose visualization is helpful in examining the possible existence of different mega-environments [50, 51]. The lines dividing the biplot into sectors represent a set of hypothetical environments. A polygon is drawn by joining all the genotypes which are located away from the biplot origin. A perpendicular line is drawn from the origin of the biplot, extending beyond the polygon, where the biplot is divided into several sectors. The genotypes located at the vertices in each sector were the best performing genotype in different environments among other genotypes in the same sector. If a genotype located on an angular vertex of the polygon falls into one sector with an environment marker (or several markers), the mean performance of this genotype would be the highest in this particular environment. Another important feature of this biplot is that it indicates environmental groupings, which suggests the possible existence of different mega-environments. Figs 3A and 4A represent a which-won-where pattern for six genotypes across three locations for pod yield and shelling percent, respectively. For pod yield, the genotype JL 501 performed better in Deesa, while NRCGCS 446 performed better in Mohanpur, West Bengal, and GG 7 performed better in Junagadh. For shelling percent, the genotypes TG 37A and TAG 24 performed better in Deesa and Mohanpur, respectively, whereas JL 501 performed better in Junagadh. Such differential ranking of genotypes across different environments suggests the possible existence of crossover GEI which is conformity with previous works [52–59].

The GGE biplot ranks the genotypes based on their mean performance and stability across environments (Figs 3B and 4B). The axis of the "average environment coordination (AEC) abscissa" is denoted by a single arrowed line that passes through the biplot origin with an arrow, which points to higher mean values for the genotypes (the best performing genotypes; high-dormancy genotypes). Therefore, the AEC ordinate is perpendicular to the AEC abscissa.

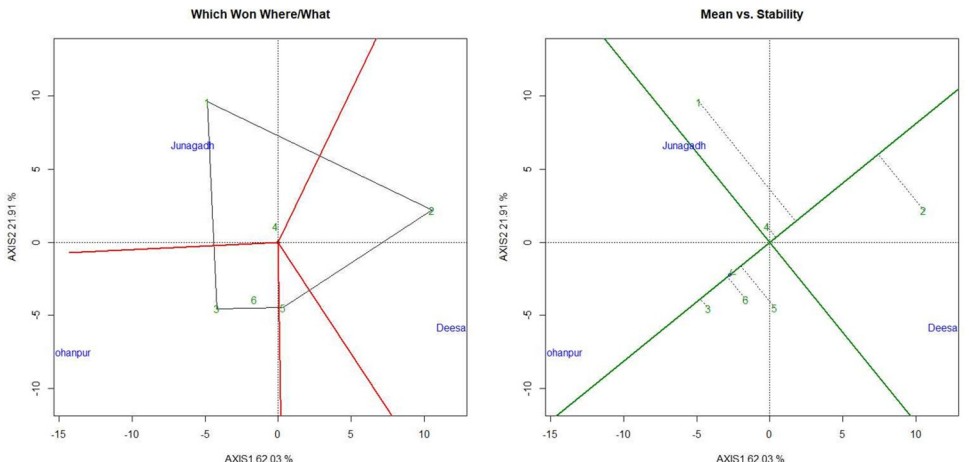

**Fig 3. GGE biplot of pod yield per plant showing the "which-won-where" pattern and mean vs. stability, indicating the rankings of genotypes (1 = GG7, 2 = JL 501, 3 = NRCGCS 446, 4 = NRCGCS 254, 5 = TG 37A, and 6 = TAG 24).**

The AEC ordinate approximates the genotypes' contributions to the G×E interaction, indicating that genotypes closer to AEC abscissa, are more consistent/ stable over environments [51, 60]. Genotype NRCGCS 446 was stable, with a high yield, whereas TAG 24 had high shelling percent and TG 37A was stable in terms of shelling percent.

## Conclusion

Evaluation of groundnut genotypes under the potato-fallow system in different states revealed differential responses of genotypes on pod yield and also higher pod yield under the potato-fallow system compared to that under the non-potato fallow system. The study identified two stable early-maturing high-yielding breeding lines, NRCGCS 254 and NRCGCS 446, which can be released for commercial cultivation and used further as potential donors for breeding improved groundnut varieties in the potato-fallow system.

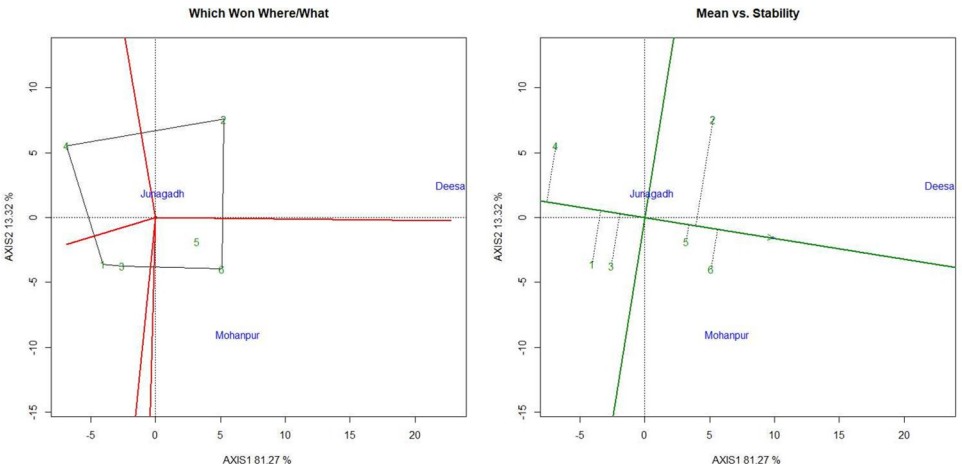

**Fig 4. GGE biplot of shelling percentage showing the "which-won-where" pattern and mean vs. stability, indicating the rankings of genotypes (1 = GG7, 2 = JL 501, 3 = NRCGCS 446, 4 = NRCGCS 254, 5 = TG 37A, and 6 = TAG 24).**

## Supporting information

**S1 Table. Performance of genotypes for maturity, SCMR and yield related characters at Junagadh over three seasons.**
(DOCX)

## Acknowledgments

Authors acknowledge the support and fascilities received from the Director, ICAR-DGR, Junagah, Gujarat.

## Author Contributions

**Conceptualization:** Gangadhara K, Ajay BC, S. K. Bera.

**Data curation:** Gangadhara K.

**Formal analysis:** Gangadhara K, Kirti Rani, Narendra Kumar.

**Investigation:** Ajay BC, Praveen Kona, Kirti Rani, Narendra Kumar.

**Methodology:** S. K. Bera.

**Software:** Ajay BC.

**Supervision:** Gangadhara K, Praveen Kona.

**Validation:** Ajay BC.

**Writing – original draft:** Gangadhara K, Ajay BC, S. K. Bera.

**Writing – review & editing:** Ajay BC, Praveen Kona, Kirti Rani, Narendra Kumar, S. K. Bera.

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
