## [Decision Letter · Decision Letter 0]

1 Nov 2022

PONE-D-22-26579Performance of some early-maturing groundnut genotypes and selection of high-yielding genotypes in the potato-fallow systemPLOS ONE

Dear Authors

Thank you for submitting your manuscript to PLOS ONE. After careful consideration, we feel that it has merit but does not fully meet PLOS ONE’s publication criteria as it currently stands. Therefore, we invite you to submit a revised version of the manuscript that addresses the points raised during the review process.

ACADEMIC EDITOR:1) Hypothesis of the study are not clear.2) Manuscript is not written technically sound, and the data do not fully support the conclusions Please ensure that your decision is justified on PLOS ONE’s publication criteria and not, for example, on novelty or perceived impact.

We look forward to receiving your revised manuscript.

Kind regards,

Jiban Shrestha

Academic Editor

PLOS ONE

Journal Requirements:

"No."

5. We note you have included a table to which you do not refer in the text of your manuscript. Please ensure that you refer to Table 7 in your text; if accepted, production will need this reference to link the reader to the Table.

Additional Editor Comments:

Authors need to include all reviewers suggestions and re-submit.

Reviewers' comments:

Reviewer's Responses to Questions

**Comments to the Author**

1. Is the manuscript technically sound, and do the data support the conclusions?

Reviewer #1: No

Reviewer #2: Yes

Reviewer #3: No

2. Has the statistical analysis been performed appropriately and rigorously? 

Reviewer #1: No

Reviewer #2: Yes

Reviewer #3: No

3. Have the authors made all data underlying the findings in their manuscript fully available?

Reviewer #1: No

Reviewer #2: Yes

Reviewer #3: Yes

4. Is the manuscript presented in an intelligible fashion and written in standard English?

Reviewer #1: No

Reviewer #2: Yes

Reviewer #3: Yes

5. Review Comments to the Author

Reviewer #1: I am not happy with the current form of the manuscript for considering to publish in PLoS One. Particulary hypothesis and novelty of the study are not clear. Also the title of the manuscript missmatch with the content of the article.

Reviewer #2: Kindly incorporate the below current Reference to safeguard the present results in a better ways, please.

Ali S, Ahmad R, Hassan MF, Ibrar D, Iqbal MS, Naveed MS, Arsalan M, Rehman A, Hussain T (2022). Groundnut genotypes’ diversity assessment for yield and oil quality traits through multivariate analysis. SABRAO J. Breed. Genet. 54(3): 565-573. http://doi.org/10.54910/sabrao2022.54.3.9

Carvalho, M.J., N. Vorasoot, N. Puppala, A. Muitia and S. Jogloy. 2018. Effects of terminal drought on growth, yield and yield components in valencia peanut genotypes. SABRAO J. Breed. Genet. 49(3) 270-279.

Girdthai T, Jogloy S, Vorasoot N, Akkasaeng C, Wongkaew S, Patanothai A, Holbrook CC (2012). Inheritance of the physiological traits for drought resistance under terminal drought conditions and genotypic correlations with agronomic traits in peanut. SABRAO J. Breed. Genet. 44(2): 240-262.

Junjittakarn J, S. Jogloy, N. Vorasoot and N. Jongrungklang (2016). Effect of mid-season drought and recovery on physiological traits and root system in peanut genotypes (Arachis hypogaea L.). SABRAO J. Breed. Genet. 48(3): 318-331.

Junjittakarn J, S. Jogloy, N. Vorasoot and N. Jongrungklang (2016). Root responses and relationship to pod yield in difference peanut genotypes (Arachis hypogaea L.) under mid-season drought. SABRAO J. Breed. Genet. 48(3): 332-343.

Kasno A and Trustinah (2015). Genotype-environment interaction analysis of peanut in Indonesia. SABRAO J. Breed. Genet. 47(4): 482-492.

Koolachart R, Jogloy S, Vorasoot N, Wongkaew S, Holbrook CC, Jongrungklang N, Kesmala T, Suriharn B (2019). Association of aflatoxin contamination and root traits of peanut genotypes under terminal drought. SABRAO J. Breed. Genet. 51(3): 234-251.

Mahakosee S, Jogloy S, Vorasoot N, Suriharn B, Puppala N, Patanothai A (2015). Genotypic diversity of traits related to nitrogen fixation in Valencia peanut germplasm. SABRAO J. Breed. Genet. 47(4): 493-501.

Singkham N, Jogloy S, Kesmala T, Swatsitang P, Jaisil P, Puppala N, Patanothai A (2011). Combining ability for oleic acid in peanut (Arachis hypogaea L.). SABRAO J. Breed. Genet. 43(1) 59–72.

Reviewer #3: The study's objective was to assess performance and adaptability, nine short duration groundnut cultivars in the potato-fallow system across three environments. This type of study requires an extensive season*location or multiple locations data to provide meaningful recommendations. The current study has on 3 environments derived from the 3 seasons which makes it insufficient. The second major flaw is the data analysis, for example in Table 2, the degrees of freedom column is not representative of the study factors, I am also in awe of the contents of the second row in the same table. The same can be said about Table 5.

6. PLOS authors have the option to publish the peer review history of their article (what does this mean?). If published, this will include your full peer review and any attached files.

Reviewer #1: No

Reviewer #2: **Yes: **PROF. DR. NAQIB Ullah KHAN

Reviewer #3: No

---

## [Author Response · Author response to Decision Letter 0]

10 Feb 2023

Response to reviewers points has been attached as a separate word file

---

## [Decision Letter · Decision Letter 1]

15 Feb 2023

Performance of some early-maturing groundnut genotypes and selection of high-yielding genotypes in the potato-fallow system

PONE-D-22-26579R1

Dear Dr. Bera,

We’re pleased to inform you that your manuscript has been judged scientifically suitable for publication and will be formally accepted for publication once it meets all outstanding technical requirements.

Kind regards,

Jiban Shrestha

Academic Editor

PLOS ONE

Additional Editor Comments (optional):

Reviewers' comments:

Reviewer's Responses to Questions

**Comments to the Author**

1. If the authors have adequately addressed your comments raised in a previous round of review and you feel that this manuscript is now acceptable for publication, you may indicate that here to bypass the “Comments to the Author” section, enter your conflict of interest statement in the “Confidential to Editor” section, and submit your "Accept" recommendation.

Reviewer #1: All comments have been addressed

Reviewer #2: All comments have been addressed

2. Is the manuscript technically sound, and do the data support the conclusions?

Reviewer #1: Yes

Reviewer #2: Yes

3. Has the statistical analysis been performed appropriately and rigorously? 

Reviewer #1: Yes

Reviewer #2: Yes

4. Have the authors made all data underlying the findings in their manuscript fully available?

Reviewer #1: Yes

Reviewer #2: Yes

5. Is the manuscript presented in an intelligible fashion and written in standard English?

Reviewer #1: Yes

Reviewer #2: Yes

6. Review Comments to the Author

Reviewer #1: I am happy to the current revised form of the article, since authors have reoslved all of my suggestions. Therefore, my final opinion is may be published the current version.

Reviewer #2: The Article is well improved in the revised version, and hence it can be accepted for publication, please...

7. PLOS authors have the option to publish the peer review history of their article (what does this mean?). If published, this will include your full peer review and any attached files.

Reviewer #1: **Yes: **Akbar Hossain, BWMRI, Bangladesh

Reviewer #2: **Yes: **Prof. Naqib Ullah Khan

---

## [Editor Report · Acceptance letter]

7 Mar 2023

PONE-D-22-26579R1 

Performance of some early-maturing groundnut *(Arachis hypogaea L.)* genotypes and selection of high-yielding genotypes in the potato-fallow system 

Dear Dr. Bera:

I'm pleased to inform you that your manuscript has been deemed suitable for publication in PLOS ONE. Congratulations! Your manuscript is now with our production department. 

Kind regards, 

on behalf of

Dr. Jiban Shrestha 

Academic Editor

PLOS ONE